# Combined Inhibition of Polo-Like Kinase-1 and Wee1 as a New Therapeutic Strategy to Induce Apoptotic Cell Death in Neoplastic Mast Cells

**DOI:** 10.3390/cancers14030738

**Published:** 2022-01-31

**Authors:** Manuela Mancini, Cecilia Monaldi, Sara De Santis, Michela Rondoni, Cristina Papayannidis, Chiara Sartor, Antonio Curti, Samantha Bruno, Michele Cavo, Simona Soverini

**Affiliations:** 1Istituto di Ematologia “Seràgnoli”, IRCCS Azienda Ospedaliero-Universitaria di Bologna, 40138 Bologna, Italy; cristina.papayannidis@unibo.it (C.P.); chiara.sartor2@unibo.it (C.S.); antonio.curti2@unibo.it (A.C.); michele.cavo@unibo.it (M.C.); 2Dipartimento di Medicina Specialistica, Diagnostica e Sperimentale Università di Bologna, 40138 Bologna, Italy; cecilia.monaldi2@unibo.it (C.M.); sara.desantis9@unibo.it (S.D.S.); samantha.bruno2@unibo.it (S.B.); simona.soverini@unibo.it (S.S.); 3Ospedale Santa Maria delle Croci, 48121 Ravenna, Italy; michela.rondoni@unibo.it

**Keywords:** systemic mastocytosis, Aurora kinase A, Polo-like kinase 1, WEE1

## Abstract

**Simple Summary:**

Systemic Mastocytosis (SM) is a rare disease resulting from a clonal proliferation of mast cells. Despite the rarity of the disease, its advanced forms remain a hard core to manage in clinical practice. At this time, there are few therapeutic options available, such as midostaurin and avapritinib, hence the need to perform a deeper investigation of the pathogenetic mechanisms involved in the development of the most aggressive forms of the disease, in order to provide further tools for the therapeutic management of this critical subset of patients. In this study, we aimed to identify new biological players involved in the pathogenesis of SM, and to evaluate their role as potential therapeutic targets. Our results identify, for the first time, two druggable markers of aggressiveness of the disease: Aurora kinase A and Polo-like kinase 1. Inhibition of these two ser/thr kinases, alone or together with Wee1, induces apoptotic cell death in SM cell lines; therefore, it appears an attractive therapeutic strategy to kill neoplastic MCs. Repurposing Polo-like kinase 1 or Aurora kinase A ± Wee1 inhibitors in advanced clinical developments for other indications is a therapeutic strategy worthy of being explored, to attempt to improve the outcome of patients with advanced SM.

**Abstract:**

Systemic mastocytosis (SM) is due to the pathologic accumulation of neoplastic mast cells in one or more extracutaneous organ(s). Although midostaurin, a multikinase inhibitor active against both wild-type and D816V-mutated KIT, improves organ damage and symptoms, a proportion of patients relapse or have resistant disease. It is well known that Aurora kinase A (AKA) over-expression promotes tumorigenesis, but its role in the pathogenesis of systemic mastocytosis (SM) has not yet been investigated. Evidence from the literature suggests that AKA may confer cancer cell chemo-resistance, inhibit p53, and enhance Polo-like kinase 1 (Plk1), CDK1, and cyclin B1 to promote cell cycle progression. In this study, we aimed to investigate the pathogenetic role of AKA and Plk1 in the advanced forms of SM. We demonstrate here, for the first time, that SM cell lines display hyper-phosphorylated AKA and Plk1. Danusertib (Aurora kinase inhibitor) and volasertib (Plk1 inhibitor) inhibited growth and induced apoptotic cell death in HMC-1.1 and -1.2 cells. Their growth-inhibitory effects were associated with cell cycle arrest and the activation of apoptosis. Cell cycle arrest was associated with increased levels of phospho-Wee1. Wee1 inhibition by MK1775 after 24 h treatment with danusertib or volasertib, when cells were arrested in G2 phase and Wee1, was overexpressed and hyper-activated, resulting in a significantly higher rate of apoptosis than that obtained from concomitant treatment with danusertib or volasertib + MK1775 for 48 h. In conclusion, Plk1 and AKA, alone or together with Wee1, are attractive therapeutic targets in neoplastic MCs. Repurposing Plk1 or AKA ± Wee1 inhibitors in advanced clinical development for other indications is a therapeutic strategy worthy of being explored, in order to improve the outcome of patients with advanced SM.

## 1. Introduction

Systemic mastocytosis (SM) is a rare disease characterized by abnormal growth and the accumulation of neoplastic mast cells (MC) in different organs, including the bone marrow (BM), with or without skin involvement. According to the World Health Organization classification, SM can be distinguished into indolent systemic mastocytosis (ISM), smoldering SM, SM with an associated hematologic neoplasm (SM-AHN), aggressive SM (ASM), mast cell leukemia (MCL), and MC sarcoma [1,2,3].

All forms of SM are characterized by variable clinical presentation and course. However, advanced-SM (adv-SM) patients (i.e., those with SM-AHN, ASM or MCL) generally show poor response to conventional drugs, and their prognosis is bad. The KIT tyrosine kinase receptor, expressed in MC progenitors as well as in mature MCs, is regarded as the major disease driver and a target in neoplastic MCs [4,5,6]. Nevertheless, KIT constitutive activation is thought to be necessary, but not enough for the development of ASM and MCL. In fact, the effects of KIT-targeting drugs, such as midostaurin, showed significant cytotoxic effects on the in vitro growth of neoplastic MC, but evident and long-lasting effects in vivo failed to be observed in some patients with adv-SM. Thus, despite the recent approval of the tyrosine kinase inhibitor midostaurin, there is still a need for target therapies for patients who relapse or show resistant disease. More recently, the KIT D816V-targeting drug avapritinib has been designed for clinical use as a single-drug treatment in AdvSM. However, only a few study data are available, and not all patients with AdvSM may respond to this kinase inhibitor. However, these studies support the efficacy of kinase inhibitors in the treatment of these malignant conditions [7]. The characterization and dissection of cooperating pathogenetic pathways may set the basis for the development of novel therapeutic strategies useful for this critical subset of patients. Moreover, a better understanding of the proliferation, survival and invasion mechanisms of normal and malignant MCs, and consequently of their vulnerabilities, will have a much wider therapeutic impact. In fact, beyond SM, MCs have recently been implicated in the pathogenesis and dissemination of several tumor types, such as thyroid, breast and colorectal cancers [8]. MCs have been found to be recruited by certain tumor cells in the surrounding stroma, where they can exert complex pro-tumorigenic roles [9]. In particular, observations in colorectal cancer have highlighted a key role for MCs in favoring a pro-inflammatory microenvironment and inducing angiogenesis [10], such that high MC density has been correlated with advanced stage and poor outcome [11]. Thus, exploring novel MC-targeting strategies holds great interest.

We have recently described, for the first time, an overexpression and hyper-activation of Polo-like kinase 1 (Plk1) and Aurora kinase A (AKA) in adv-SM, as compared to ISM, probably playing a key role in the pathogenesis and progression of SM (Mancini et al. submitted). AKA and Plk1 have a central role in the regulation of cell cycle progression and mitosis, and are frequently overexpressed in human cancers. Thus, Plk1 and AKA are considered to be attractive targets for cancer therapy in solid tumors, myeloid leukemias, and myelodysplasias [12]. Plk1 plays different roles in cell cycle progression: it controls mitotic entry and the G2/M checkpoint and regulates the centrosome formation, spindle assembly, and chromosome segregation. Moreover, it is implicated in cytokinesis and meiosis. It is essential for the regulation of cell division and genome stability in mitosis, and participates in DNA damage response [13,14,15,16,17]. The Aurora-kinase family includes serine/threonine kinases involved in mitosis regulation, and is required for the maintenance of genome stability. AKA and B regulate multiple aspects of mitosis, including centrosome duplication, formation of a bipolar mitotic spindle, and most importantly, play a key role in the spindle assembly checkpoint [18,19,20].

Since Plk-1 and AKA and B are clearly implicated in tumorigenesis, they are considered promising anticancer drug targets. The literature has already reported the overexpression and hyper-activation of Aurora kinases and Polo-like kinases in patients with CML, acute myeloid leukemia and acute lymphoblastic leukemia, highlighting that AKA and Polo-like kinase inhibitors are potential candidates for the treatment of several hematological malignancies.

Therefore, in this study, we aimed to investigate whether AKA and Plk1 inhibition may be a promising therapeutic strategy in advSM.

## 2. Results

### 2.1. Danusertib (AKA Inhibitor) and Volasertib (Plk1 Inhibitor) Inhibited Growth in HMC-1.1 and -1.2 Cells

We have already demonstrated that the serine/threonine kinases Aurora A and Plk1 are hyper-activated and over-expressed in patients with adv-SM, when compared to ISM and to a pool of healthy donors [21].

To better understand if Plk1 and AKA may play a role in the proliferation advantage of neoplastic MCs, we tested HMC-1, ROSA^KITWT^ and ROSA^KITD816V^ cell lines for AKA and Plk1 expression to assess whether they could represent adequate in vitro models to validate and further explore our in vivo results. Both HMC-1.1 and 1.2 subclones showed an overexpression and an hyperactivation of AKA and Plk1, whereas neither ROSA^KITWT^ nor ROSA^KITD816V^ showed any AKA or Plk1 hyper-activation (Appendix A). Starting from this evidence, we performed clonogenic assays using HMC-1.1 and -1.2 cells in the presence of danusertib or volasertib (AKA and Plk1 inhibitors, respectively). AKA inhibition by danusertib induced a significant dose-dependent reduction in colony formation, with LD50 ranging from 26.67 to 53.33 nM in both HMC-1.1 and -1.2 subclones, as shown in Figure 1A. Even better results were achieved by Plk1 inhibition by volasertib with a LD50 ranging from 4.83 nM to 9.67 nM (Figure 1B). A significant reduction of clonogenic activity was associated with increased apoptotic cell death (Figure 1C).

In Figure 1D we reported the IC50 (µM) calculated by using a dedicated software (Compusyn) after a dose-escalation experiment in which danusertib and volasertib treatment, for 24 h, inhibited growth and induced apoptosis in HMC-1.1 (IC50 = 0.6490 µM and 0.4437 µM, respectively) and -1.2 cells (IC50 = 0.8921 µM and 0.8078 µM, respectively).

As shown in Figure 1E, danusertib treatment exerts on the target effects by inhibiting Aurora A phosphorylation; moreover, in Figure 1F we show that Aurora A and Plk1 inhibition alone induced apoptotic cell death as a result of caspase 9 and caspase 3 cleavage.

### 2.2. Inhibition of Either AKA or Plk1 Induces Cell Cycle Arrest and Affects G2/M Checkpoint Proteins

The growth-inhibitory effects of danusertib and volasertib were found to be associated with mitotic arrest, as shown in Figure 2A. Cell cycle arrest was accompanied by increased levels of phospho (p)-Chk1(S317), p-Chk2(T68), p-cyclin B1(S133), p-CDK1(Y15) and p-Wee1(S642) (Figure 2B). All the evaluated proteins were implicated in the regulation of the G2/M checkpoint, and their hyperphosphorylation represented a clear sign of blocked cell transition into the mitotic phase.

### 2.3. Combined WEE1 and AKA, or PLK1, Inhibition Has Synergistic Effects in HMC-1 Cells

Building from the evidence that AKA and Plk1 inhibition induced an increase in p-WEE1 as a consequence of G2/M checkpoint activation, we tried to block WEE1 kinase by using MK1775 in order to force mitosis entry and induce apoptosis as a result of DNA damage propagation.

Apoptosis activation was demonstrated by an increase in annexin-V-positive cells and by the detection of the cleaved forms of caspase 3, caspase 8, caspase 9 and PARP.

WEE1 inhibition by MK1775 (500 nM) after 24 h treatment with danusertib or volasertib (100 nM), when cells were arrested in G2 and WEE1 was over-expressed and hyper-activated, resulted in a significantly higher rate of apoptosis than that observed after concomitant treatment with danusertib or volasertib (100 nM) + MK1775 (500 nM) for 48 h (Figure 3A,B). The apoptotic process was found to be activated by caspase 3, caspase 8, caspase 9 and PARP cleavage, ultimately resulting in bax release from mitochondria (Figure 3C,D). Moreover, both combinations resulted in a significant increase in the DNA double-strand break marker γH2 AX (Figure 3C,D), confirming our hypothesis that WEE1 inhibition promotes mitosis and propagates genomic instability.

As shown in Figure 3E,F, after cell line treatment with danusertib or volasertib and MK1775 for 24 + 24 h, with increasing doses of the drugs (0.25–1.0 µM), the CI (calculated with Compusyn) showed that the combination of danusertib and MK1775 had synergistic effects both in HMC-1.1 and -1.2 sub-clones (CI < 1), while the combination of volasertib and MK1775 had an additive effect in HMC-1.1 (CI = 1) and a synergistic effect in HMC-1.2 (CI < 1). Report data obtained by Compusyn software are collected in the Appendix A.

The toxicity of the drug combinations was evaluated in normal cell lines (BaF3 and 32D) by using a cell growth assay in liquid medium and an apoptotic assay. Both methods exerted superimposable results; in fact, no relevant toxic effects were observed in normal cells using both drug combinations and both schedules (Appendix A).

Finally, we decided to test the combination of midostaurin, currently approved both by the Food and Drug Administration (FDA) and by the European Medicines Agency (EMA) to treat SM, and MK1775 (WEE1 inhibitor). We had previously assessed the effects of WEE1 inhibition + midostaurin on clonogenic potential in neoplastic MCs from two patients with MCL and one patient with ASM. This approach induced a dose-dependent reduction in colony formation capacity (with LD50 ranging from 40 to 70 nM for midostaurin + MK1775). MK1775 + midostaurin combination showed synergistic effects compared to either agent alone (Combination Index, 0.76) [22]. Here, we show the effects of the drug combination in HMC-1.1 and -1.2 cell lines (Figure 4). The statistical analysis of the data obtained from apoptotic cell death assays in the HMC-1.1 cell line after treatment with MK1775 and midostaurin, alone and in combination, showed that the combination had a strong synergistic effect on inducing apoptosis, while in HMC-1.2 cells, the combination had an additive effect (Table 1 and Table 2). Our conclusions are supported by data analysis performed with Compusyn.

The combination of AKA inhibition + midostaurin was not evaluated, being biologically redundant. Indeed, midostaurin is a pan-inhibitor that, beyond KIT, includes AKA among its targets (although danusertib is a much more potent inhibitor of AKA).

## 3. Discussion

SM is a rare myeloproliferative disorder that is characterized by a heterogeneous spectrum of disease variants ranging from indolent forms displaying a near-normal life expectancy, to aggressive forms with a poor outcome [23,24]. The D816V-activating KIT gene mutation represents a criterion for SM diagnosis and plays an important pathogenetic role. However, this mutation is detectable in ≥90% of patients; therefore, this single lesion cannot explain the phenotypical and clinical heterogeneity of SM by itself [25,26,27].

We have recently observed, for the first time, that the over-expression and hyper-activation of AKA and Plk1 serine/threonine (ser-thr) kinases represents a common feature of patients with advanced SM (but not of ISM patients) [21]. We can speculate that AKA and Plk1 hyper-activity may enhance the effects of the constitutive activation of the KIT pathway, determining the aggressive phenotype of advanced SM. AKA is a kinase that has been implicated in the regulation of mitosis and is being explored as a potential target in solid tumors and in leukemias. The AKA-targeting compound danusertib has been shown to induce cell cycle arrest and apoptosis in several neoplastic models [28]. Plk1 is a ser/thr-protein kinase that has different important functions throughout the M phase of the cell cycle and is involved in the regulation of mitotic exits and cytokinesis. It has been reported to be over-expressed and hyper-activated in several solid tumors and hematologic malignancies including SM, where it has already been proposed as a potential therapeutic target [29]. In this study, we investigated whether AKA and Plk1 play a role in the proliferation advantage of neoplastic MCs and whether their inhibition may be of clinical interest. To this purpose, we screened the two main experimental SM cell line models, HMC-1 and ROSA, for AKA and Plk1 expression and activity by WB. We found that only HMC-1 cells recapitulate our observations in advSM patients. This is consistent with the fact that HMC-1 cells were established from an MCL patients, whereas ROSA cells are normal CD34+ cells from umbilical cord blood ‘engineered’ to express mutant KIT; thus, they do not fully mirror the complexity of advanced disease. Therefore, all our subsequent experiments could be performed on HMC-1 cells only.

Our findings show that danusertib and volasertib treatment, by inhibiting AKA and Plk1 activity, trigger apoptosis in a dose-dependent manner and reduce clonogenic potential by inducing an accumulation in the G2 phase of the cell cycle. In association with the block in G2 phase, we detected a clear effect on the post-translational modification of several proteins known to be involved in the G2/M transition and to be regulated by AKA or Plk1, such as cyclin B1 (phosphorylated on ser 133) and WEE1 (phosphorylated on ser 642).

Starting from the evidence and from previous knowledge that WEE1 protein kinase is over-expressed and hyper-activated in the G2 phase, we decided to test a novel concept of drug combination, danusertib or volasertib plus a WEE1 inhibitor (MK1775), with the aim to enhance apoptotic cell death in our in vitro model. We therefore decided to add MK1775 at the exact stage in the cell cycle when WEE1 is hyper-activated: after 24 h of inhibition of AKA or Plk1. The combination we devised turned out to be very efficient compared to single-drug treatments. In the HMC-1.2 subclone (carrying the major SM driver mutation, the D816V KIT-activating mutation), cytotoxicity data obtained from apoptotic cell death analysis demonstrated a synergistic activity of both danusertib + MK1775 and volasertib + MK1775. In the HMC-1.1 subclone, the danusertib + MK1775 combination showed synergistic effects and volasertib + MK1775 showed additive effects. The toxicity of the combinations was explored in normal cell lines (32D and BaF3), but no effects were observed either in survival curves, in which drug combination induced only a slight reduction in growth, and in apoptotic assay in which no relevant toxic effect was observed, either in BaF3, or in the 32D cell lines.

To further explore the clinical value of combination therapies in advSM, we also evaluated the addition of midostaurin, the current standard of treatment. Midostaurin has a broad inhibition spectrum, with many kinases, beyond KIT, among its targets (for example, being also a Flt3 inhibitor, it is used also in acute myeloid leukemia). AKA is one such kinase [30], although danusertib is much more potent than midostaurin. We thus reasoned that the most rational strategy to combine midostaurin was with a WEE1 inhibitor. In line with our previous findings in primary neoplastic MCs from advSM patients [22], we indeed observed a strong synergistic effect in HMC-1.1 cells and an additive effect in HMC-1.2 cells.

Our mechanistic studies about the pathways involved in apoptotic cell death activation provide an explanation as to why AKA and Plk1 inhibition, alone or in combination with WEE1 inhibition, may induce apoptosis in MCL cell lines. We hereby propose a model that demonstrates the roles of AKA and Plk1 over-expression and hyper-activation in signaling pathways operating in SM cells, and the effects of their inhibition. In such a model, both AKA and Plk1 have important roles in activating proliferation and uncontrolled cell division.

In conclusion, our data suggest that the combined targeting of AKA or Plk1 + WEE1 may be an excellent strategy to target neoplastic MCs and pave the way for exploring a new opportunity for the salvage therapy of advSM. It will be interesting to assess whether our findings may enable the therapeutic targeting of malignant MCs in other neoplastic conditions, where they are known to play an important role in tumor growth and progression [31].

## 4. Materials and Methods

### 4.1. Cell Lines

The two subclones of the HMC-1 MC leukemia cell line (HMC-1.1, which harbors the KIT V560G but not the KIT D816V mutation, and HMC-1.2, which harbors both activating mutations) were maintained in Iscove Modified Dulbecco Medium (IMDM) medium, additioned with 10% fetal calf serum, 1% l-Glutamineand 100 µM penicillin, 100 µg/mL streptomycin (Gibco, Thermo Fisher Scientific, Waltham, MA, USA), in 5% CO_2_ and a fully humidified atmosphere at 37 °C. Every three days, cells were pelleted at 1200 rpm for 5 min and the culture medium was changed. The cells were resuspended at a concentration of 2 × 10^5^ cells/mL [32]. The ROSA^KITWT^ and ROSA^KITD816V^ cell line, obtained by transfecting CD34^+^ umbilical cord blood cells with KIT WT or mutant (D816V) were cultured in KIT ligand stem cell factor (SCF)-containing medium or in SCF free medium (IMDM supplemented with 1% penicillin/streptomycin, 1% sodium pyruvate, 1% MEM vitamins, 2% MEM non-essential amino acids, 1% of L-glutamine, 1% commercial solution of insulin transferrin-sodium selenite, 0.3% bovine serum albumin, all purchased by GIBCO), respectively. These were kindly provided by Prof. Michel Arock (Department of Hematological Biology, Pitié-Salpêtrière Hospital, Pierre et Marie Curie University, Paris). Moreover, 32D and BaF3 cell lines, used as normal controls cell lines, were maintained at 37 °C in RPMI 1640 medium additioned with 10% FCS, and 10% WEHI-3 conditioned medium as source of IL-3 and antibiotics.

### 4.2. Drug Treatments

Danusertib (Aurora kinases inhibitor), volasertib (Plk-1 inhibitor), MK1775 (WEE1 inhibitor) and midostaurin (pan-kinase inhibitor; all purchased from Selleckchem, Houston, TX, USA) were used to investigate the effects of AKA, Plk1, Wee1 and KIT inhibition in HMC-1.1 and HMC-1.2 cells. The biological consequences on cell cycle distribution and apoptotic cell death activation of the aforementioned drugs were evaluated after 24- and 48-h treatment of each compound alone, or in combination, (100 nM of danusertib and volasertib and 500 nM of MK1775). Cell cycle distribution of HMC-1.2 cell line untreated and treated with 100 nM danusertib for 24 hours by flow cytometry can be found in Appendix A. 

Apoptosis was assessed by measuring the uptake of fluoresceinated Annexin V and propidium iodide (PI) (F. Hoffmann-La Roche Ltd., Basel, Switzerland) according to published methods [33]. A FACsCantoII flow cytometer (Beckton Dickinson, Franklin Lakes, NJ, USA) set at 488 nm excitation and 530 nm wavelength bandpass filter for fluorescein detection or 580 nm for PI detection, and a dedicated software (DIVA software, Beckton Dickinson) were used. Data obtained from cytotoxicity assays were analyzed by using the CompuSyn software (ComboSyn, Inc.; Paramus, NJ, USA) in order to calculate the efficacy of drugs, alone or in combination. Combination indexes were calculated using the same software [34]. Briefly, sub-lethal drug doses were established to perform further experiments and to assess drugs doses inducing early apoptotic cell death, but not necrotic cell death. To this purpose, we used a drug dose escalation starting from 0.25 µM to 1 µM of each drug, and in combination. After 24 h, treatment cells were evaluated for Annexin V uptake and the percentage of residual living cells was used to calculate IC50 values by using a dedicated software (Compusyn).

Cytofluorimetric analysis of cell cycle distribution was performed by assessing the uptake of propidium iodide (PI) (Roche) (488 nm excitation wavelength and 530 nm bandpass filter wave for fluorescein and >580 nm for PI detection) and analyzing the results with the Cell Quest software (Becton Dickinson).

Drug efficacy was evaluated in clonogenic assays. In HMC-1.1 and HMC-1.2, reduction of colony (generated in 0.9% methylcellulose supplemented with 30% fetal calf serum) number in the presence of increasing doses of danusertib (25–100 nM) and volasertib (10–50 nM) was assessed after 14 days of incubation at 37 °C in a fully humidified atmosphere and 5% CO2. Nonlinear regression analysis (GraphPad Prism; GraphPad Software Inc., San Diego, CA, USA) was employed to calculate the lethal dose (LD50) of the drugs, alone and in combination [22].

Drug toxicity was evaluated in normal cells (BaF3 and 32D) by using a growth curve assay. Briefly, cells were maintained under control conditions or treated with danusertib or volasertib (100 nM) + MK1775 (500 nM) for 24, 48, 72 and 96 h. At each timepoint, cells were stained with trypan blue and counted by using a Burker chamber in order to discriminate living cells from dead cells.

### 4.3. Protein Analysis

Whole cell lysates were used for protein analyses by Western blot (WB), immunoblotting (IP), and the evaluation of histone post-translational modifications according to published methods [35].

The anti-caspase 3 (cat. n. 9662), anti-caspase-8 (cat. n. 9748), anti-caspase 9 (cat.n. 9502), anti-Bax (cat. n. 5023), anti-phospho-CHK1 (S317) (cat. n. 12302), anti-phospho-CHK2 (T68) (cat. n. 2197), anti-phospho-cyclin B1 (S133) (cat. n. 4133), anti-phospho-WEE1 (S642) (cat. n. 4910), anti-phospho-CDK1 (Y15) (cat. n. 4539), anti-phospho-H2 AX (S139) (cat. n. 2577), anti-phospho-Aurora kinase A (T288) (cat. n. 3079), anti-phospho-Plk1 (T210) (cat. n. 5472) and anti-PARP (cat. n. 5625) antibodies were purchased from Cell Signaling Technology (Danvers, MA, USA). The anti-β-actin antibody (cat. n. sc-8432) used as a loading control was purchased from Santa Cruz Biotechnology (Dallas, TX, USA). Signal intensities in single blots from three separate experiments were acquired using a ChemiDoc XRS+ Gel Imaging System (BioRad, Hercules Contra Costa County, Hercules, CA, USA) equipped with a dedicated software (Image Lab, BioRad, Hercules, CA, USA).

### 4.4. Statistics

Data are presented as mean values ± SD. Data were analyzed for statistical significance by the Student t-test using the GraphPad Prism Software. A *p* value of <0.05 was considered as statistically significant.

## 5. Conclusions

Our findings open new therapeutic avenues in adv-SM. Besides allogeneic transplant, cladribine, interferon, or the recently approved KIT inhibitor, midostaurin are currently the main option for advSM patients, but long-term disease management may be difficult in several patients who fail or cannot tolerate these therapies. Thus, other treatment options are urgently needed. Our results show that targeting the AKA/Plk1/WEE1 axis may be an effective strategy in SM. AKA, Plk1 and Wee1 have been extensively explored as drug targets in several neoplastic conditions; the main advantage of such a strategy would be that drugs already approved or in advanced clinical development for other conditions could easily be repurposed for use in advSM. This would streamline the introduction of novel therapeutic approaches in a rare disease such as advSM.

## Figures and Tables

**Figure 1 cancers-14-00738-f001:**
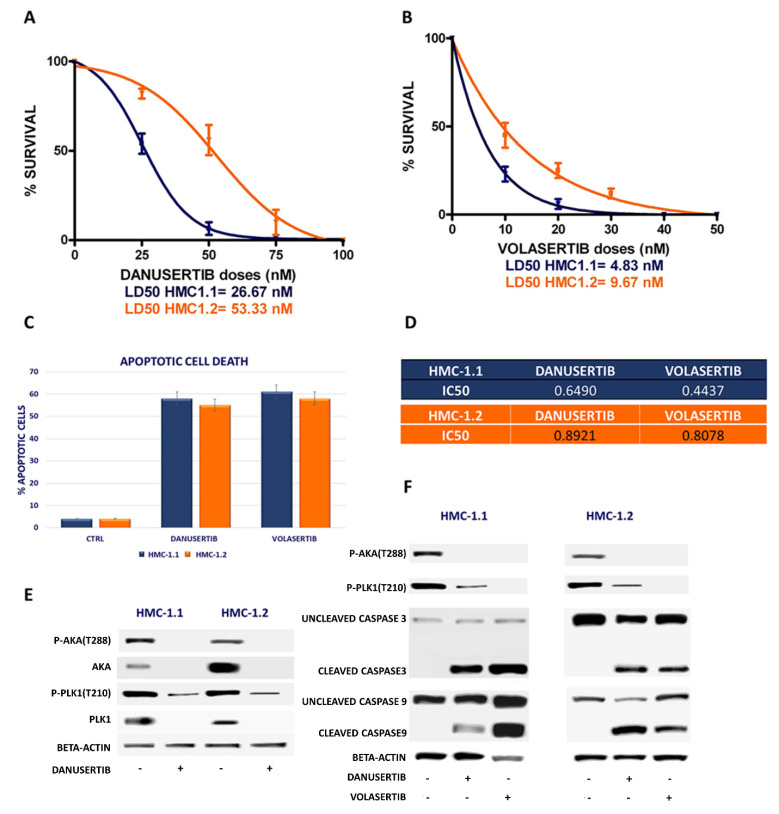
Effects of AKA and Plk1 inhibition on HMC-1 cell line proliferation. (**A**,**B**) Dose-dependent inhibition of clonogenic capacity induced by danusertib and volasertib treatment in HMC-1.1 (blue lines) and HMC-1.2 cells (orange lines), evaluated by 14-day methylcellulose colony-forming assay. Cells were treated with increasing concentrations of danusertib (AKA inhibitor) and volasertib (Plk1 inhibitor). (**C**) Flow cytometry analysis of Annexin-V/PI positive cells. HMC-1.1 and HMC-1.2 cells were treated with danusertib (100 nM) and volasertib (100 nM) for 24 h. (**D**) A dose-escalation experiment was performed to calculate the IC50 of danusertib and volasertib after 24 h treatment. (**E**) Western blot analysis was performed to test AKA and Plk1 expression and activity and to confirm the on-target effects of danusertib treatment. (**F**) Western blot analysis of apoptosis-related proteins in HMC-1.1 and HMC-1.2 cells treated with 100 nM danusertib and volasertib for 24 h. Beta-actin was used as a loading control. Clonogenic assays in HMC-1.1 and HMC-1.2 cells were repeated three times in an independent way: results were represented as a mean of three biological replicates. Western blotting assays were performed starting from lysates obtained by three independent drug treatments for each sample. Full Western blot images can be found in Appendix A.

**Figure 2 cancers-14-00738-f002:**
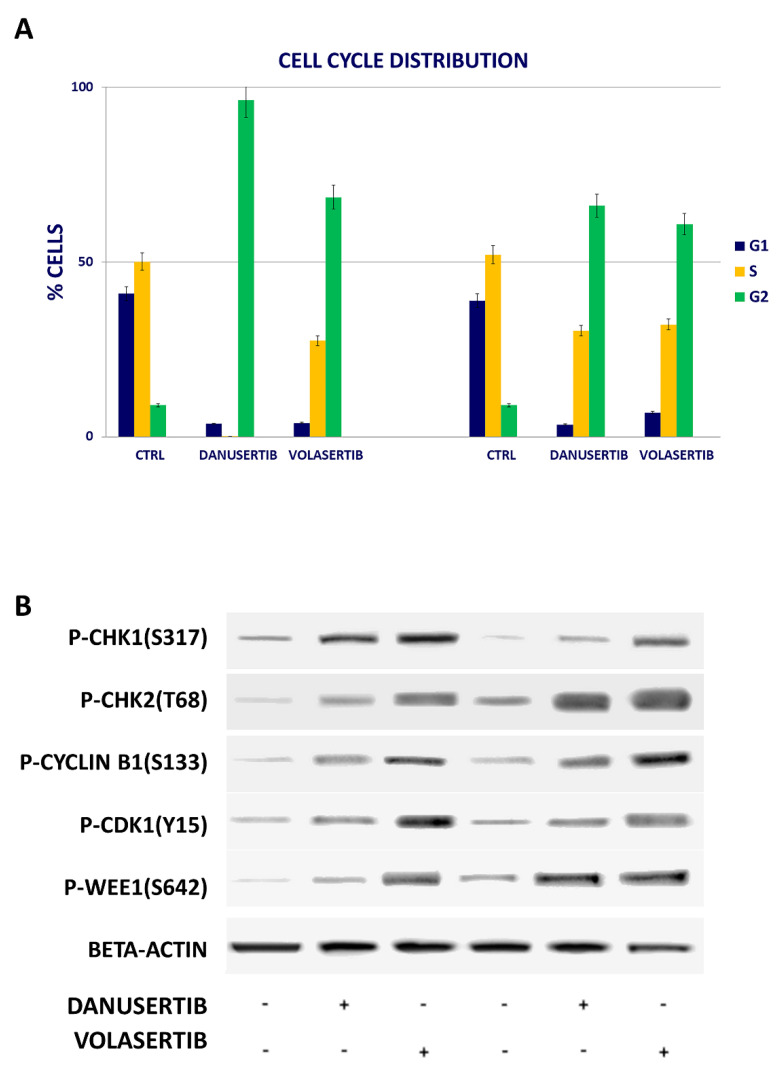
Effects of AKA and Plk1 inhibition on cell cycle. (**A**) Flow cytometry analysis of cell cycle was performed by using HMC-1.1 and HMC-1.2 cell lines treated with 100 nM danusertib or volasertib for 24 h. (**B**) Western blot analysis of cell cycle-related proteins in HMC-1.1 and HMC-1.2 treated with 100 nM danusertib and volasertib for 24 h (the sign + indicates samples treated with above mentioned drugs, while the sign – indicates samples cultured without the mentioned drug). Beta-actin was used as loading control. Cell cycle distribution assays were represented as a mean of three biological replicates. Similarly, all Western blotting assays were performed starting from lysates obtained by three independent drug treatments for each sample.

**Figure 3 cancers-14-00738-f003:**
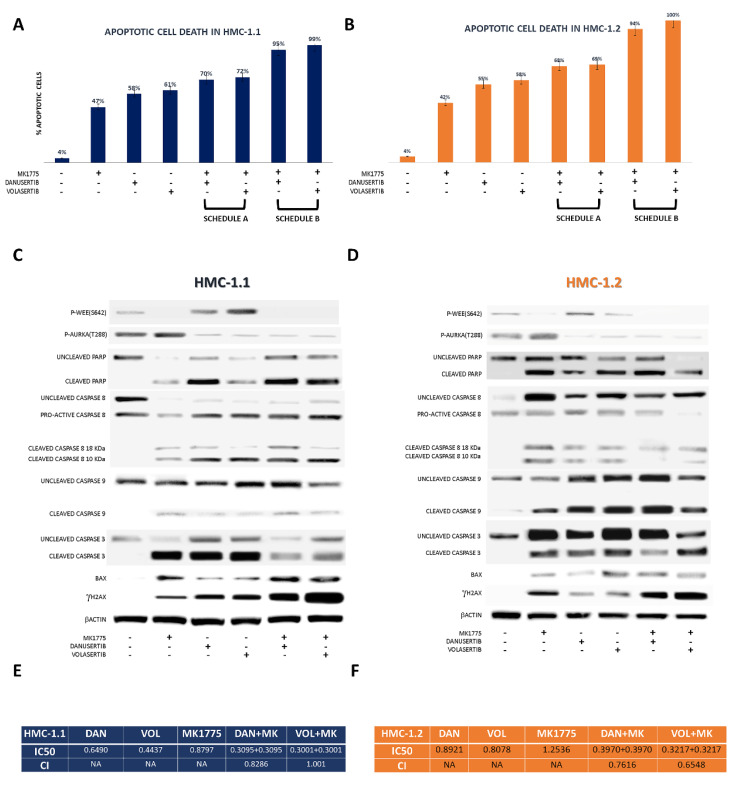
Effects of AKA or Plk1 inhibition in association with WEE1 inhibition. Flow cytometry analysis of apoptosis induction in HMC-1.1 (**A**), and HMC-1.2 (**B**) cells, following 48 h treatment with danusertib or volasertib alone or in combination with MK1775 (schedule A, 48 h with both drugs), or 24 h-treatment with danusertib or volasertib followed by 24 h-treatment with danusertib or volasertib + MK1775 (schedule B). Western blot analysis of apoptosis-related proteins in HMC-1.1 (**C**) and HMC-1.2 (**D**) cells following treatment with AKA, Plk1 or WEE1 inhibitors as single agents or in combination according to schedule B. Beta-actin was used as a loading control. (**E**,**F**) Results obtained by cell death data analysis performed by using Compusyn software and CI calculation. Apoptotic cell death evaluations were represented as a mean of three biological replicates. Similarly, all Western blotting assays were performed starting from lysates obtained by three independent drug treatments for each sample.

**Figure 4 cancers-14-00738-f004:**
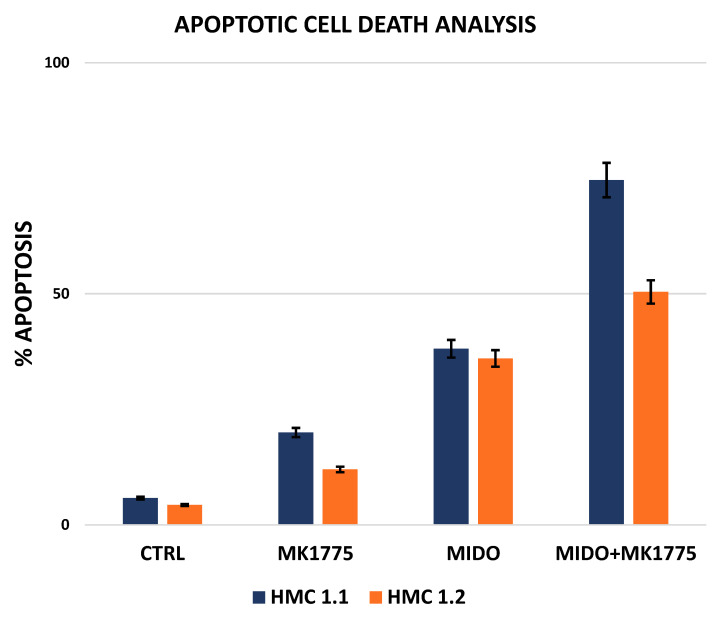
Effects of WEE1 or KIT inhibition alone or in combination. Flow cytometry analysis of apoptosis induction in HMC-1.1 (**blue**) and HMC-1.2 (**orange**) cells following 24 h treatment with MK1775 or midostaurin (1 µM) alone or in combination. Columns represent the mean of three independent experiments and the bars represent the standard error. ctrl, control.

**Table 1 cancers-14-00738-t001:** HMC-1.1. Effects of WEE1 or KIT inhibition alone or in combination. Data obtained by cytofluorimetric evaluation of apoptotic cell death were analyzed by using a dedicated software (Compusyn) and the combination index (CI) was calculated. Fa is the fraction of cells affected by drugs treatment.

DRUGS	Dose (µM)	Fa	CI
AZD1775	1	0.20	NA
MIDOSTAURIN	1	0.38	NA
AZD1775 + MIDOSTAURIN	1 + 1	0.76	0.172

**Table 2 cancers-14-00738-t002:** HMC-1.2. Effects of WEE1 or KIT inhibition alone or in combination. Data obtained by cytofluorimetric evaluation of apoptotic cell death were analyzed by using a dedicated software (Compusyn) and the combination index (CI) was calculated. Fa is the fraction of cells affected by drugs treatment.

DRUGS	Dose (µM)	Fa	CI
AZD1775	1	0.12	NA
MIDOSTAURIN	1	0.36	NA
AZD1775 + MIDOSTAURIN	1 + 1	0.50	0.905

## Data Availability

Part of the supporting data are included in the Appendix A. Others available upon request.

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
