# Peer review of "Combined Inhibition of Polo-Like Kinase-1 and Wee1 as a New Therapeutic Strategy to Induce Apoptotic Cell Death in Neoplastic Mast Cells"

_cancers, 2022, doi:10.3390/cancers14030738_

Round 1

Reviewer 1 Report

The manuscript can be accepted

Author Response

Thank you for your evaluation of our manuscript.

Reviewer 2 Report

The authors did not answer to my main request, which was to analyse what are the effects of their combination on normal cells and/or other cell lines. Moreover, their answers to the majority of my other comments are not satisfactory...

For example, concerning AKA and Plk1 overexpressions/hyperactivations, data that are not showed here, they are arguing that it has already been reported in an abstract 10.1182/blood-2018-99-113278 that is not peer-reviewed and the figure on that specific point is not publicly available.

Also, their answer to my comment about testing their “combination with one of the available drug to treat systemic mastocytosis, such as Midostaurin” is disconcerting: they already tested the combination Midostaurin + WEE1 inhibitor in a previous study, and they did not make any effort to add a work about it in the introduction, for example…

Etc, etc…

Author Response

We thank Reviewer 2, as well as the Academic Editor, for their additional comments and requests aimed to further improve our manuscript. We have revised the text accordingly and added Supplementary Figures, in order to satisfy all their requests. The new changes to the text are highlighted in green.

Response to Reviewer

The authors did not answer to my main request, which was to analyse what are the effects of their combination on normal cells and/or other cell lines. Moreover, their answers to the majority of my other comments are not satisfactory...

We apologize. The reason why other SM cell lines could not be tested had been explained to another Reviewer and we failed to repeat it. Please find this explanation below. We indeed had initially omitted in the text that, in an attempt to identify a cell line model where to further investigate our initial observations in advSM patients and where to test novel pharmacological approaches, both HMC-1 cells and ROSA cells had been screened for AKA and Plk1 expression and activation. We had found that AKA and Plk1 are overexpressed and hyperactivated in HMC-1.1 and -1.2 cells, but not in ROSAKITWT or ROSAKITD816V. Thus, only HMC-1 cells recapitulated our observations in advSM patients. This was not surprising: it is consistent with the fact that HMC-1 cells were established from an MCL patients, whereas ROSA cells are normal CD34+ cells from umbilical cord blood ‘engineered’ to express mutant KIT, thus they do not fully mirror the complexity of advanced disease. This is why all our experiments could be performed in HMC-1 cells only. This has now been clearly reported in the Results section and discussed in the Discussion section. A supplementary Figure (Figure S1) with supporting information has been created.

To address Reviewer’s request to analyse the effects of our drug combinations in normal cells, we have now performed cell growth curves in liquid medium using both BaF3 and 32D cell lines, and this has now been reported in the Results section. The Ba/F3 cell line is an IL-3-dependent mouse pro-B cell line derived from BALB/c mouse strain. The 32D differs from BaF3 because it belongs to the myeloid and not the lymphoid differentiation lineage. Preliminary Western blot analysis was performed to exclude AKA overexpression and hyperactivitation in both cell lines and results are now shown in the supplementary section (Figure S1). To perform cell growth curves and test drugs combination in normal cells, BaF3 and 32D cell lines were maintained under control conditions or treated with danusertib or volasertib (100nM) + MK1775 (500 nM) for 24, 48, 72 and 96 hours. At each timepoint, cells were stained with trypan blue and counted by using a Burker chamber in order to discriminate living cells from dead cells and only the living cells were considered to perform the growth curves. As can be seen from Figures S2B and S2C both the combination of danusertib and MK1775 and the combination of volasertib and MK1775 do not show any cytotoxic effects, although the growth of treated cells is obviously slower if compared with that of cells cultured in control medium (Figure S2A). Finally, apoptotic cell death was assessed by measuring the uptake of fluoresceinated Annexin V and propidium iodide (PI), results are shown in Supplementary section (Figure S3).

For example, concerning AKA and Plk1 overexpressions/hyperactivations, data that are not showed here, they are arguing that it has already been reported in an abstract 10.1182/blood-2018-99-113278 that is not peer-reviewed and the figure on that specific point is not publicly available.

As we had proposed in our previous reply, we are now here confidentially sharing with this Reviewer the Western blotting image, although a comprehensive manuscript where we not only report this finding but also extensively investigate the pathogenetic implications of AKA and Plk1 hyperactivation for the development of advSM is submitted for publication. We have now introduced in the text the reference to the ASH abstract reporting this finding, that is publicly available. This same Western blotting image appeared on the poster that was publicly presented at the meeting. We understand Reviewer’s perplexity about the fact that we cannot show this image in the present manuscript, but presenting the same results in two separate manuscripts would be unfair.

Also, their answer to my comment about testing their “combination with one of the available drug to treat systemic mastocytosis, such as Midostaurin” is disconcerting: they already tested the combination Midostaurin + WEE1 inhibitor in a previous study, and they did not make any effort to add a work about it in the introduction, for example…

We apologize for the misunderstanding. We have now retested the combination of midostaurin and MK1775 (WEE1 inhibitor) as suggested and results have been introduced in the text. The effects of the drug combination in HMC-1.1 and 1.2 cell lines were measured by using a cytofluorimetric assay, and statistical analysis of data obtained from apoptotic cell death evaluation was performed by using Compusyn. Our results are now displayed in the Results section (page 7 lines 1-5; page 8 lines 1-9; Tables 1 and 2 and Figure 4). The combination of midostaurin and AKA/Plk1 inhibition was not tested because biologically redundant (midostaurin has a broad inhibition spectrum, with many kinases, beyond KIT, among its targets, and AKA is one such kinases, although danusertib is much more potent than midostaurin). This has now been clearly explained and discussed in the text.

Round 2

Reviewer 2 Report

The authors tried to answer my comments as much as they could, but unfortunately, I persist in thinking that as long as their other paper they are referring to in the introduction is not peer-reviewed/published, this article is incomplete, lacking to demonstrate the importance of their research project.

“We have recently described for the first time an overexpression and hyper-activation of Polo-like kinase 1 (Plk1) and Aurora kinase A (AKA) in adv-SM as compared to ISM, probably playing a key role in the pathogenesis and progression of SM (Mancini et al. submitted).”

Moreover, they are unable to reproduce any of their results in other cell lines, thus leading to minimise the impact of their conclusions

Minor comments:

- This sentence in the simple summary does not make sense and should be changed: “Despite the rarity of the disease, its advanced forms remain a hard core to manage in clinical practice”.

- All molecular weight markers are missing in Western blot figures (main and supplemental).

This manuscript is a resubmission of an earlier submission. The following is a list of the peer review reports and author responses from that submission.

Round 1

Reviewer 1 Report

Overall, the study is well-designed and reported, and I believe the observations obtained will add significant knowledge to the community. Overall, it is a sound body of work, the paper is well-written but I have only 1 minor comment which will help strengthen the manuscript:

Information (such as cat. no, and the company) of the all the antibodies used in the study must be included in the methods section.

Author Response

Reviewer 1

Overall, the study is well-designed and reported, and I believe the observations obtained will add significant knowledge to the community. Overall, it is a sound body of work, the paper is well-written but I have only 1 minor comment which will help strengthen the manuscript:

Information (such as cat. no, and the company) of the all the antibodies used in the study must be included in the methods section.

The methods section has been amended according to this Reviewer’s remark regarding all detailed information about the antibodies used (page 8 section 4.3)

Reviewer 2 Report

Systemic mastocytosis is a dangerous and rare myeloid neoplasm whose treatment is based on only few available drugs, rarely achieving significant disease control.

In this study Mancini et al. aimed to investigate the pathogenetic role of Aurora kinase A and Plk1 in the advanced forms of SM. It was demonstrated here for the first time that primary neoplastic MCs and SM cell lines display hyper-phosphorylated Aurora kinase A and Plk1. The results of the experiments performed in vitro in two currently used mastocytosis cell lines support that Plk1 and Aurora kinase A, alone or together with Wee1, are attractive therapeutic targets in neoplastic MCs. The results reported in this study are original and potentially interesting for the development of new therapeutic strategies in systemic mastocytosis.

Comments/Criticisms

  1. The authors should mention in the introduction the recent approval of Avapritinib for the treatment of systemic mastocytosis as single-drug treatment. These studies support the efficacy of kinase inhibitors in the treatment of this malignant condition.
  2. This study is based on two cell lines, HMC-1.1 and HMC-1.2 cells. However, the use of additional cell lines, such as ROSAKITWT, ROSAKITD816V and MCPV-1 would consistently improve the preclinical impact of the findings reported in this study.
  3. The analysis of the effect of the two drugs in the in vivo ROSAKITD816V model would be important to further improve the preclinical significance of this study.

Author Response

Reviewer 2

  • The authors should mention in the introduction the recent approval of Avapritinib for the treatment of systemic mastocytosis as single-drug treatment. These studies support the efficacy of kinase inhibitors in the treatment of this malignant condition.

The introduction has been amended according to the Reviewer’s comment regarding detailed information about recent approval of Avapritinib in clinical practice (page 2 lines 14-18).

  • This study is based on two cell lines, HMC-1.1 and HMC-1.2 cells. However, the use of additional cell lines, such as ROSAKITWT, ROSAKITD816V and MCPV-1 would consistently improve the preclinical impact of the findings reported in this study.

  • The analysis of the effect of the two drugs in the in vivo ROSAKITD816V model would be important to further improve the preclinical significance of this study.

We agree with this Reviewer on the fact that, normally, the use of a wider panel of cell lines would consistently improve the soundness and impact of preclinical findings.

This is why, following our work in HMC-1.1 and -1.2 cells, we tested ROSAKITWT and ROSAKITD816V cell lines for AKA and Plk1 expression to assess whether they could represent an additional model where to validate our results. Unfortunately, we found that ROSAKITD816V do not show any AKA or Plk1 hyper-activation, as demonstrated in the figure below. We assumed that the lack of AKA and Plk1 reflects the different way the cell lines were established. While HMC-1 cells derive from a MCL patient, ROSA cells were generated by transfecting cord blood-derived normal MC progenitor cells with a construct coding for KITD816V. AKA/Plk1 axis is not activated downstream of the KIT receptor, but is rather a cooperating event in advSM. This was the reason why no further experiments using these cell lines were performed.

MCPV-1 cells are not yet available in our laboratory, nor is it possible to purchase them. However, since the MCPV-1 cell line was obtained by lentiviral immortalization of cord blood-derived MC progenitor cells, similarly to ROSA cells, they are likely to not display AKA/Plk1 deregulation, either.

Reviewer 3 Report

This is an original article regarding a new therapeutic strategy to induce apoptotic cell death in neoplastic mast cells.

I have the following comments:

The should be consistent with the use of abbreviations. I suggest to report Aurora Kinase A with an abbreviation.

The study is really interesting and well developed. 

The clinical implications of the study must be added by introducing the role mast cells in the different tumors (at least in the introduction)

Mastocytosis, MCAS, and Related Disorders-Diagnosis, Classification, and Therapy. Int J Mol Sci. 2021 May 10;22(9):5024. doi: 10.3390/ijms22095024

Mast Cells, microRNAs and Others: The Role of Translational Research on Colorectal Cancer in the Forthcoming Era of Precision Medicine. J Clin Med. 2020 Sep 3;9(9):2852. doi: 10.3390/jcm9092852. PMID: 32899322; PMCID: PMC7564551.

Methods section should be clarified (samples, rather than period of the study). 

The period of the study is mandatory in the methods section and also the limitations. 

Author Response

Reviewer 3

The should be consistent with the use of abbreviations. I suggest to report Aurora Kinase A with an abbreviation.

All the manuscript has been amended according to this Reviewer’s suggestion and Aurora kinase A has been replaced with AKA throughout the text.

The study is really interesting and well developed. 

The clinical implications of the study must be added by introducing the role mast cells in the different tumors (at least in the introduction)

Mastocytosis, MCAS, and Related Disorders-Diagnosis, Classification, and Therapy. Int J Mol Sci. 2021 May 10;22(9):5024. doi: 10.3390/ijms22095024

Mast Cells, microRNAs and Others: The Role of Translational Research on Colorectal Cancer in the Forthcoming Era of Precision Medicine. J Clin Med. 2020 Sep 3;9(9):2852. doi: 10.3390/jcm9092852. PMID: 32899322; PMCID: PMC7564551.

We thank this Reviewer for this interesting suggestion, aimed to enhance the clinical relevance and translation of our findings possibly beyond SM. We have now modified the Introduction (page 2, lines 23-33) and Discussion section (page 7, lines 51-52, page 8 lines1-2) introducing the key role of MCs in favouring the development and progression of several tumors (with relevant references to the literature) and the interest of providing novel therapeutic opportunities for MC targeting, respectively.

Methods section should be clarified (samples, rather than period of the study). 

The period of the study is mandatory in the methods section and also the limitations.

We thank the reviewer for the observation that allows us to clarify a misunderstanding caused by the description in the abstract section and in the introduction of results obtained from experiment carried out in primary cells from SM patients. These observations has been previously reported in an abstract presented at the 60th ASH meeting (Manuela Mancini et al. MDM2 and Aurora Kinase a Contribute to SETD2 Loss of Function in Advanced Systemic Mastocytosis: Implications for Pathogenesis and Treatment. Blood (2018) 132 (Supplement 1): 1779.), and a comprehensive manuscript where we not only report this finding but also extensively investigate the pathogenetic implications of AKA and Plk1 hyperactivation for the development of advSM is submitted for publication. Therefore, it is not applicable to this type of study, performed only in SM cell lines, the definition of a period in which we recruited patient samples for our investigation.

Reviewer 4 Report

Mastocytosis is a rare disorder for which there is still no first-line standard therapy available for patients with advanced systemic mastocytosis (the most severe form). Here, the authors are developing a new therapeutic strategy based on the inhibition of a mitotic kinase, Plk1 or Aurora Kinase A, in combination with the inhibition of a key regulator of the cell cycle, the Wee1 kinase.

Unfortunately, I have several major concerns that must be addressed.

Major comments:

The aim of this paper is to define whether targeting Plk1 or AKA, which are found overexpressed/hyperactived in patients with advanced systemic mastocytosis, can be a potential therapeutic strategy. However, these data are not publicly available yet. It is then difficult to judge the importance and the impact of this study. Moreover, no analysis on overexpression/hyperactivation of these targets were conducted in HMC1.1 and HMC1.2 cell lines. These data are however essential and must be performed in comparison to normal mast cells and various other cell lines.

Also, both HMC-1.1 and HMC-1.2 are responsive to all treatments. It is then necessary to determine how normal cells and other cell lines will respond to such inhibitor combination.

While on the subject of combinations, it is very difficult to understand why the authors did not perform any assays in combination with one of the available drug to treat systemic mastocytosis, such as Midostaurin…

Minor comments:

- The authors wrote in the abstract that HMC1 are “primary neoplastic MCs and SM cell lines”. This sentence is incorrect and misleading: HMC-1.1 and HMC-1.2 are sister cell lines.

- On almost all Western-blots, contrasts were worked out too strongly and should then be improved.

- In Figure 1 A-B, the authors indicate that the clonogenic capacity was evaluated after 14-days, while in the materials & methods it is indicated 10-days. Please correct it. Moreover, colony-forming assays are not a survival assay; the legend on Figure 1 A and B graphs should then be amended.

- In Figure 1D, more details should be given on the dose-escalation experiment that was used to calculate the IC50, and the associated graphs should be shown. Moreover, was it really a dose-escalation assay that was performed? Or a cell growth inhibition assay? Is a dose-escalation not an in vivo assay? That point should also be clarified, and the materials and methods updated.

- Annotations on Figure 1 E-F are difficult to read.

- Western-blots showing the impact of danusertib and/or volasertib on the activation states of Plk1 and AKA must be added in Figure 1 E and F.

- Representative cell cycle graphs with all gates should be shown for Figure 2 A.

- Protein quantification should be added in Figure 2 B.

- As requested for Figure 1D, graphic data should be provided for Figure 3 E-F.

Author Response

Reviewer 4

Major comments:

The aim of this paper is to define whether targeting Plk1 or AKA, which are found overexpressed/hyperactived in patients with advanced systemic mastocytosis, can be a potential therapeutic strategy. However, these data are not publicly available yet. It is then difficult to judge the importance and the impact of this study. Moreover, no analysis on overexpression/hyperactivation of these targets were conducted in HMC1.1 and HMC1.2 cell lines. These data are however essential and must be performed in comparison to normal mast cells and various other cell lines.

Also, both HMC-1.1 and HMC-1.2 are responsive to all treatments. It is then necessary to determine how normal cells and other cell lines will respond to such inhibitor combination.

We thank the reviewer's comments that gave us the opportunity to make the purpose of our study clearer. We have reported that AKA and Plk1 are over-expressed and hyper-activated in advSM in an abstract presented at the 60th ASH meeting (Manuela Mancini et al. MDM2 and Aurora Kinase a Contribute to SETD2 Loss of Function in Advanced Systemic Mastocytosis: Implications for Pathogenesis and Treatment. Blood (2018) 132 (Supplement 1): 1779.), and a comprehensive manuscript where we not only report this finding but also extensively investigate the pathogenetic implications of AKA and Plk1 hyperactivation for the development of advSM is submitted for publication. Briefly, we checked AKA and Plk1 expression and phosphorylation status in primary cells from patients with ISM (n=11) and ASM (n=14). Both AKA and Plk1 were found to be overexpressed and activated in patients with advSM (14/14) as compared to patients with ISM, who displayed protein and phosphorylation levels superimposable to those of healthy donors. Supporting Western blotting images can be confidentially shared with this Reviewer upon request.

While on the subject of combinations, it is very difficult to understand why the authors did not perform any assays in combination with one of the available drug to treat systemic mastocytosis, such as Midostaurin.

We understand this Reviewer’s concern. The combination of midostauring + WEE1 inhibition has already been explored in a previous published work by our group (Martinelli, G. et al. SETD2 and histone H3 lysine 36 methylation deficiency in advanced systemic mastocytosis «LEUKEMIA», 2018, 32, pp. 139 – 148) We assessed the effects of WEE1 inhibition+midostaurin on clonogenic potential in neoplastic MCs from two patients with MCL and one patient with ASM. This approach induced a dose-dependent reduction in colony formation capacity (with LD50 ranging from 40 to 70nM for midostaurin+MK1775). MK1775+midostaurin combination showed synergistic effects as compared to either agent alone (Combination Index, 0.76). The combination of AKA inhibition + midostaurin is, in our opinion, biologically redundant, since midostaurin is a pan-inhibitor that includes AKA among its targets (although danusertib is a much more potent inhibitor of AKA).

Minor comments:

- The authors wrote in the abstract that HMC1 are “primary neoplastic MCs and SM cell lines”. This sentence is incorrect and misleading: HMC-1.1 and HMC-1.2 are sister cell lines.

We agree with this Reviewer on the fact that the sentence “primary neoplastic MCs and SM cell lines” is incorrect, therefore the abstract was amended according reviewer suggestion.

- On almost all Western-blots, contrasts were worked out too strongly and should then be improved.

The acquisition of western blot experiments was performed by using the ChemiDoc XRS+ Gel Imaging System (Biorad) using a software that automatically sets the timing of image acquisition in order to optimize the brightness and contrast of the image. In order to calculate densitometry using ImageJ software, we maintain the high contrast recommended by thesoftware. No other changes have been performed to the published images and most importantly every change made was the same for the entire gel analysed.

If the reviewer considers the contrast recommended by the software too high, we will correct the images keeping the original contrast

- In Figure 1 A-B, the authors indicate that the clonogenic capacity was evaluated after 14-days, while in the materials & methods it is indicated 10-days. Please correct it. Moreover, colony-forming assays are not a survival assay; the legend on Figure 1 A and B graphs should then be amended.

The manuscript was amended according the reviewer comment, the number of days required to generate visible colonies for the HMC-1 cell line is 14 days, I apologise for the mistake.

- In Figure 1D, more details should be given on the dose-escalation experiment that was used to calculate the IC50, and the associated graphs should be shown. Moreover, was it really a dose-escalation assay that was performed? Or a cell growth inhibition assay? Is a dose-escalation not an in vivo assay? That point should also be clarified, and the materials and methods updated.

We thank the reviewer for this comment that gave us the opportunity to clarify the rationale of our experiment. We decided to perform a dose-escalation experiment to define the drug doses required to induce sub-lethal effects in our cell lines. For this purpose, each drug, alone or in combination, was added at scalar doses into the cell culture medium, starting from 0.25 to 1 µmolar. After 24 hours treatment cells were evaluated for Annexin V uptake and the percentage of residual living cells was used to calculate IC50 values by using a dedicated software (Compusyn).

Compusyn reports are shown below and in the Supplementary section.

Compusyn reports were showed below and in the Supplementary section.

Data for Drug: MK1775 [µM] in HMC-1.2 cell line

Dose (µM)

Effect (% of living cells)

0.25

0.92

0.5

0.87

0.75

0.69

1.0

0.55

4 data points entered.

Data for Drug: volasertib [µM]

Dose (µM)

Effect (% of living cells)

0.25

0.85

0.5

0.68

0.75

0.51

1.0

0.43

4 data points entered.

Data for Drug: danusertib [µM]

Dose (µM)

Effect (% of living cells)

0.25

0.88

0.5

0.72

0.75

0.53

1.0

0.48

4 data points entered.

Data for Drug: MK1775 [µM] in HMC-1.1 cell line

Dose (µM)

Effect (% of living cells)

0.25

0.98

0.5

0.92

0.75

0.70

1.0

0.58

4 data points entered.

Data for Drug: volasertib [µM]

Dose (µM)

Effect (% of living cells)

0.25

0.91

0.5

0.72

0.75

0.61

1.0

0.48

4 data points entered.

Data for Drug: danusertib [µM]

Dose (µM)

Effect (% of living cells)

0.25

0.81

0.5

0.68

0.75

0.47

1.0

0.35

4 data points entered.

- Annotations on Figure 1 E-F are difficult to read.

Figures 1E and 1D were amended according to reviewer’s suggestions.

- Western-blots showing the impact of danusertib and/or volasertib on the activation states of Plk1 and AKA must be added in Figure 1 E and F.

Western blotting showing Plk1 and AKA dephosphorylation after danusertib or volasertib treatment were added in figure 1F as suggested, while had already been shown in figure 1E.

- Representative cell cycle graphs with all gates should be shown for Figure 2 A.

A representative cell cycle graph is shown below and inserted in the supplementary section. It indicates a flow cytometry evaluation of cell cycle distribution of HMC-1.2 cell line untreated and treated with 100 nM danusertib for 24 hours. Cell cycle analysis was performed excluding the sub-G1 population, including dead cells, and polyploid cells.

- Protein quantification should be added in Figure 2 B.

An Excel file showing the protein quantification of all western blots performed in the study, will be uploaded during resubmission.

- As requested for Figure 1D, graphic data should be provided for Figure 3 E-F.

All requested data were included in the Compusyn report shown below regarding the combination effects of danusertib and MK1775 or volasertib and MK1775.

HMC1.2

Data for Drug Combo: (MK1775+danusertib [1:1])

Dose (µM)

Effect (% of living cells)

0.25

0.62

0.5

0.54

0.75

0.27

1.0

0.13

4 data points entered.

Total Dose (µM)

Effect (% of living cells)

CI Value

0.5

0.62

0.65277

1.0

0.54

1.06118

1.5

0.27

0.76950

2.0

0.13

0.58012

Data for Drug Combo: (MK1775+volasertib [1:1])

Dose (µM)

Effect (% of living cells)

0.25

0.60

0.5

0.51

0.75

0.19

1.0

0.10

4 data points entered.

Total Dose (µM)

Effect (% of living cells)

CI Value

0.5

0.60

0.50261

1.0

0.51

1.00001

1.5

0.19

0.65120

2.0

0.10

0.42010

HMC-1.1

Data for Drug Combo: (MK1775+danusertib [1:1])

Dose (µM)

Effect (% of living cells)

0.25

0.57

0.5

0.49

0.75

0.21

1.0

0.09

4 data points entered.

Total Dose (µM)

Effect (% of living cells)

CI Value

0.5

0.57

0.61423

1.0

0.49

1.01253

1.5

0.21

0.82475

2.0

0.09

0.52140

Data for Drug Combo: (MK1775+volasertib [1:1])

Dose (µM)

Effect (% of living cells)

0.25

0.60

0.5

0.51

0.75

0.19

1.0

0.10

4 data points entered.

Total Dose (µM)

Effect (% of living cells)

CI Value

0.5

0.60

0.50261

1.0

0.51

1.00001

1.5

0.19

0.82456

2.0

0.10

0.42010
